# Glenohumeral Osteoarthritis: A Biological Advantage or a Missed Diagnosis?

**DOI:** 10.3390/jcm13082341

**Published:** 2024-04-18

**Authors:** Camille Crane, Caleb Wagner, Stephen Wong, Bryce Hall, Jillian Hull, Katharine Irwin, Kaitlin Williams, Amanda Brooks

**Affiliations:** 1Colorado Campus, Rocky Vista University College of Osteopathic Medicine, 8401 South Chambers Road, Greenwood Village, CO 80112, USA; bryce.hall@co.rvu.edu (B.H.); katharine.irwin@rvu.edu (K.I.);; 2Utah Campus, Rocky Vista University College of Osteopathic Medicine, 855 East Center Street, Ivins, UT 84738, USA; calebwagner95@gmail.com (C.W.); stephen.wong@ut.rvu.edu (S.W.); jillian.nicholas@ut.rvu.edu (J.H.); abrooks@rvu.edu (A.B.)

**Keywords:** shoulder osteoarthritis, glenohumeral joint, glenohumeral osteoarthritis, glenohumeral osteoarthritis prevalence, glenohumeral osteoarthritis incidence, shoulder osteoarthritis diagnostic classification system, glenohumeral osteoarthritis radiographic classification

## Abstract

(1) **Background:** Osteoarthritis is a degenerative joint disease that is commonly diagnosed in the aging population. Interestingly, the lower extremity joints have a higher published incidence of osteoarthritis than the upper extremity joints. Although much is known about the disease process, it remains unclear why some joints are more affected than others. (2) **Methods:** A comprehensive literature review was conducted utilizing the search engines PubMed, Google Scholar, and Elsevier from 2014 to 2024, directing our search to osteoarthritis of various joints, with the focus being on glenohumeral osteoarthritis. (3) **Results and Discussion:** The literature review revealed a publication difference, which may be explained by the inconsistency in classification systems utilized in the diagnosis of shoulder osteoarthritis. For instance, there are six classification systems employed in the diagnosis of glenohumeral osteoarthritis, making the true incidence and, therefore, the prevalence unobtainable. Furthermore, susceptibility to osteoarthritis in various joints is complicated by factors such as joint anatomy, weight-bearing status, and prior injuries to the joint. (4) **Conclusions:** This review reveals the lack of understanding of shoulder osteoarthritis’s true incidence and prevalence while considering the anatomy and biomechanics of the glenohumeral joint. In addition, this is the first paper to suggest a single criterion for the diagnosis of glenohumeral osteoarthritis.

## 1. Introduction

Osteoarthritis (OA) is characterized by the gradual breakdown of cartilage in the joint, resulting in various health consequences, especially in older individuals (Figure 1). According to findings from the Third National Health and Nutrition Examination Survey, approximately 37% of adults in the United States who are 60 years of age or older have radiographic evidence of OA [1]. Considering an aging population, the significant consequences of OA are increasingly relevant; the progressive nature of osteoarthritis results in functional deterioration and disability, which drives the socioeconomic healthcare burden. This burden is profound, affecting various aspects of individuals’ lives and society. Healthcare costs related to OA, including expenses for medication, therapy, and surgeries, exemplify the financial burden associated with managing this condition [2]. Additionally, decreased productivity due to pain and mobility limitations affects both employees and employers, leading to increased absence and reduced efficiency in the workplace. In severe cases, OA can result in disability and unemployment, imposing additional financial strain and possible career adjustments [3]. Caregivers supporting individuals with OA also encounter socioeconomic challenges as they balance caregiving responsibilities with work commitments. Addressing the socioeconomic impact of OA necessitates comprehensive strategies encompassing healthcare access, preventive measures, and workplace accommodations, ultimately enhancing the well-being of affected individuals and society at large. The Disability-Health survey in France suggests that osteoarthritis is a major contributor to daily activity limitation, such that 22% reported walking difficulties, 19% struggled with carrying objects, and 13% faced challenges in getting dressed. While a study from a single country is unlikely to be representative of all patient populations, it can provide a reasonable groundwork for epidemiological data in disease states such as OA. Much of the epidemiological data on OA consist of smaller data samples from subpopulations; therefore, the reported figures are simply estimates. Furthermore, the reliance on immediate family (9%), health professionals (12%), and delivery health services (9%) emphasizes the substantial impact on independent living [4]. While the exact pathogenesis of OA remains unknown, there are known risk factors that contribute to its development, including obesity, sedentary lifestyle, increasing age, genetics, diet, previous joint injury, abnormal joint loading, and the female sex [1,4,5,6,7] (Figure 2). In addition to risk factors, OA development and progression are associated with multi-tissue pathologies involving cartilage, subchondral bone, and synovium. Even before there are visible changes in subchondral bone, histological changes observed in OA-affected synovium encompass various abnormalities such as synovial lining hyperplasia, infiltration of macrophages and lymphocytes, neoangiogenesis, and fibrosis [8,9,10]. Furthermore, several upregulated genes (i.e., CCL3, CHST11, GPR22, PRKAR2B, and PTGS2) that have elevated expression in osteoarthritic glenoid cartilage have been identified as being associated with OA [11], as have several biomarkers [12]. Synovial inflammation, observed in all stages of OA, has been linked to pain and joint dysfunction [13]. Current evidence suggests that the inflammatory mechanism driving OA progression includes Toll-like receptor engagement and complements cascade activation, promoting extracellular matrix degradation of joint tissues. This process also triggers the production and release of damaging cytokines and chemokines that are detrimental to chondrocytes [8].

Furthermore, proteinases have been linked to impair the structural components of the cartilage extracellular matrix (ECM), exacerbating chondrocyte degradation [14]. Additionally, chondrocytes undergo microenvironmental metabolic changes in response to environmental stress, likely contributing to OA phenotype characteristics [15].

This review postulates reasons for the lower incidence of glenohumeral OA (GHOA) compared to other joints. It also discusses how the biomechanics of the upper extremity and lower extremity joints impact OA. Lastly, this review proposes the need for future research to increase awareness and improve patient outcomes and treatment options.

## 2. Methods

A comprehensive literature review was conducted utilizing the search engines PubMed, Google Scholar, and Elsevier. Key search phrases included “shoulder osteoarthritis”, “glenohumeral osteoarthritis”, “osteoarthritis”, “osteoarthritis incidence”, “osteoarthritis prevalence”, and “osteoarthritis diagnosis.” The initial search yielded 652 articles from the period between 2014 and 2024. Included in this review are articles written in English focused on the incidence, diagnosis, and management within the United States healthcare framework. Excluded from consideration were studies focusing on particular surgical methods, outcomes metrics, and glenohumeral osteoarthritis arising as a consequence of surgical intervention or rotator cuff injuries. To aid the comprehension of clinically relevant aspects of glenohumeral osteoarthritis (GHOA), the authors consulted resources such as UpToDate and Orthobullets. Figures were crafted using BioRender, and explicit permissions were sought and obtained for the inclusion of any proprietary material utilized in the figures.

## 3. Results and Discussion

The prevalence of OA differs between the upper and lower extremities [16]. The Global Burden of Disease study describes a higher prevalence of OA in lower extremity joints (knee and hip) compared to upper extremity joints (shoulder and hand). This study ranked the prevalence (from most to least) of OA as follows: knee, hip, and then hand [16]; however, other sources state that GHOA is the third most common large joint type of OA [5]. It is noteworthy that this review agrees with the order of joint OA prevalence suggested in the study by the Global Burden of Disease. Ansok and Muh estimated that between 16% and 20% of adults older than 65 years of age in the United States show radiographic evidence of GHOA [17]. Their findings were based on previously published data. In their 2020 “Management of Glenohumeral Osteoarthritis: Evidence Based Clinical Practice Guidelines”, the American Academy of Osteopathic Surgeons stated that the true global prevalence and incidence of GHOA cannot be currently estimated. They do, however, report that radiological data have found the prevalence of GHOA to be 94% in women and 85% in males over 80 years old, as well as a 20% incidence of idiopathic GHOA in adults older than 60 presenting for shoulder symptoms [18]. Additionally, Baumann et al. similarly estimated the prevalence of GHOA in individuals over the age of 65 to be 20% [19]. Moreover, global epidemiological studies provide well-defined incidence rates of OA in the knee [20], hip [21], and hand [22], and yet, the incidence of GHOA is inconclusive [5,18,23,24,25] (Figure 3). Incidence and prevalence can vary significantly among subpopulations, and as such, these figures are simply estimates and cannot accurately predict the incidence and prevalence of GHOA among all patient populations.

Some studies suggest that upper extremity joints, like the elbow and wrist, have a lower prevalence of osteoarthritis than GHOA. Elbow OA ranges from 2% to 3% in prevalence, while wrist arthritis (including rheumatoid arthritis, post-traumatic arthritis, and osteoarthritis) is approximately 14% [26,27]. Both elbow and wrist OA are typically observed in older adults, although either can occur in younger individuals. The manifestation of OA in younger patients may be a result of repetitive movements, prior ligament tears, or previous fractures [4]. Notably, shoulder osteoarthritis is often seen in individuals with previous shoulder injuries (e.g., dislocations or rotator cuff tears), overuse of the joint, certain occupations, or repetitive overhead sports [5,17,28]. Overall, it is established that the knee joint has the highest prevalence of OA compared to all other joints.

Knee OA affects women more frequently than males, particularly after menopause [4] (Figure 2), whereas hip OA affects women and men equally [29]. The occurrence of lower extremity OA is influenced by factors such as geographical location, ethnicity, and population-specific variables [30]. Still, it is crucial to acknowledge that the onset of OA can, either in lower extremity or upper extremity joints, vary over time due to dynamic factors like shifts in lifestyle patterns and population demographics. Other factors that contribute to differences are the distinct anatomical structures of the joints with unique ranges of motion and the individual diagnostic pathways for each joint.

This prompts the question: Does the lower incidence of glenohumeral osteoarthritis stem from the anatomy, biomechanics, and physiological function of the glenohumeral joint, or is it due to diagnostic challenges?

The lower incidence of glenohumeral OA could be attributed to several factors related to the anatomy, biomechanics, and physiological function of the glenohumeral joint. Unlike weight-bearing joints like the knee and hip, which are subjected to significant mechanical stress and wear over time, the glenohumeral joint experiences relatively less load-bearing activity [31]. Additionally, the glenohumeral joint has a wide range of motion. It is stabilized by a complex network of ligaments, tendons, and muscles, which may contribute to its ability to withstand degenerative changes [10]. Furthermore, the presence of synovial fluid within the joint cavity provides lubrication and nourishment to the articular cartilage, potentially mitigating the development of OA [10].

However, it is also important to consider diagnostic challenges as a contributing factor to the perceived lower incidence of glenohumeral OA. Diagnosis of glenohumeral OA can be challenging due to the complexity of the joint and the variability in symptom presentation [32]. Moreover, radiographic evaluation of the shoulder may be less sensitive than other joints, making it difficult to detect early signs of OA [33]. Diagnostic criteria and classification systems for shoulder OA, such as those proposed by Elsharkawi et al. and Samilson and Prieto, may also lack consensus and standardization, leading to variability in reporting and potentially underestimating the prevalence of the condition [32,34]. Therefore, while anatomical and biomechanical factors may contribute to the lower incidence of glenohumeral OA, diagnostic challenges likely play a role in the observed trends as well.

Conducting research to understand the underlying reasons behind the indeterminate true incidence and true prevalence of GHOA is crucial. We hypothesize that the unexplored biomechanical advantages or compensatory mechanisms of the shoulder, compared to the knee and hip joints, may explain this.

### 3.1. Anatomy of the Shoulder and Knee Joints

The glenohumeral joint of the shoulder is a diarthrodial synovial ball and socket joint. Stabilization consists of both static and dynamic stabilizers. The static stabilizers include the bony articulations, glenohumeral ligaments, glenoid labrum, and negative intra-articular pressure. Foremost, the bony anatomy consists primarily of the articulation between the glenoid articular surface and the humeral head [31]. The precise angles of these bony landmarks can impact the development and progression of GHOA [35,36,37,38,39,40,41]. The ligaments of the shoulder joint are the superior glenohumeral ligament, middle glenohumeral ligament, inferior glenohumeral ligament, and coracohumeral ligament. Next, the glenoid labrum assists in centering the humeral head, is an attachment point for the glenohumeral ligaments, deepens the socket in which the humeral head sits in, and acts as an anti-shear bumper, protecting the bony articulations [31]. The last static stabilizer highlighted is the negative intra-articular pressure of the glenohumeral joint, which is related to the glenoid labrum, which deepens the joint socket. The pressure within the joint is approximately −4 mmHg, creating a vacuum that helps prevent distraction or subluxation of the glenohumeral joint [42]. It has been shown that disrupting this pressure allows for frequent subluxation [43]. The muscles surrounding the shoulder, including the scapulothoracic and rotator cuff muscles, constitute the dynamic stabilizers. The rotator cuff muscles stand out for their capability to withstand shear forces and contribute to joint stability. They achieve this by working in conjunction with capsular ligaments to safeguard the joint [31]. It is worth noting that in addition to the static and dynamic stabilizers, the acromioclavicular, scapulothoracic, and sternoclavicular joints also contribute significantly to shoulder stability [1].

As previously mentioned, a study from the Global Burden of Disease did not list GHOA in the top three most prevalent joints to develop OA; therefore, it is intriguing to speculate that the non-weight-bearing portion of the glenohumeral joint might contribute to a lower incidence of OA. Generally, a maximum of 30% of the humeral head’s cartilage contacts the glenoid articular surface at any given time. As OA advances, there is wearing of cartilage, causing the bones to come into direct contact and articulate with each other. However, the incidence of GHOA is lower compared to other joints, which cannot be solely explained by the low percentage of cartilage contact; the limited contact percentage would reduce the overall contribution to the progressive bone-on-bone contact [31,44]. Also, osteoarthritis affects weight-bearing lower extremity joints as well as non-weight-bearing upper extremity joints [45,46]; so, weight-bearing status is not the only explanation for why the glenohumeral joint may have a lower incidence compared to the hip, knee, and hand joints. The weight-bearing status of a joint is noted as a risk factor due to the increased force experienced through the joint, which may predispose the joint to develop OA in the future. Non-weight-bearing joints may still develop OA as a result of increased force experienced through the joint as a result of overuse or injury. The biomechanical forces that contribute to the development of OA will be discussed in detail later.

Since the knee joint has the highest prevalence of OA, a brief outline of the major anatomy of the knee is useful. Interestingly, the knee is also a diarthrodial synovial joint like the shoulder. The femorotibial joint is a hinge joint that involves articulation between the femur’s medial and lateral condyles and the tibia’s corresponding plateaus—this is the primary weight-bearing structure of the knee and allows flexion, extension, and internal and external rotation. The patellofemoral joint is a plane joint involving articulation between the patella and the femoral trochlea, increasing the mechanical advantage of the extensor muscles. In addition to the knee articulations, there are four primary ligaments of the knee—the anterior cruciate ligament, posterior cruciate ligament, medial collateral ligament, and lateral collateral ligament—and two menisci, namely the lateral and medial menisci [47]. Both the shoulder and the knee are diarthrodial synovial joints, with differences in ligamentous and muscular structures surrounding each; as discussed above, much is known regarding the anatomy of the shoulder and the knee joint, but there is a lack of discussion comparing and contrasting the anatomy of the joint with the biomechanics of the joint. There is a deficiency in the current research surrounding the prevalence of the shoulder joint—one may wonder whether the shoulder joint exhibits biomechanical superiority over other joints or if it is related more to diagnostic challenges. Without further research to understand the prevalence of the shoulder, it will remain unknown.

### 3.2. Joint Biomechanics

Transitioning from the examination of shoulder and knee anatomy, osteoarthritis may be linked to biomechanical factors rather than solely relying on joint anatomy. It has been proposed that variations in biomechanics, joint angles, and other factors may contribute to differences in the incidence and prevalence of joint dysfunction [48,49,50]. Still, the lack of data on the incidence and the prevalence of GHOA makes it difficult to rank the true prevalence, as previously mentioned.

Joints experience degradation either following a traumatic event or from wear and tear over a lifetime. Traumatic damage to the cartilage initiates and propagates a cascade of additional harm, eventually resulting in the development of OA [51,52]. To incur wear and tear damage, the joint surfaces must come close enough to exert force on the cartilage of the opposing bone surface. Maximum proximity of a joint surface is exhibited in a closed-pack position (CPP), indicating the highest level of surface congruency [53] (Figure 4a,c). Contrary to a CPP is a maximally loose-packed position (MLPP); the MLPP refers to the bones when they are at their maximum distance from each other, thereby reducing shear forces within the joint (Figure 4b,d). As the joint moves away from an MLPP toward a CPP, the bones of the joint draw closer together, narrowing the joint space and increasing shear forces within the joint.

The position of the joint in either a CPP or an MLPP is inherent to its normal physiological function, and its role in weight bearing becomes a key consideration. In the knee, a CPP is reached at full extension—this alignment is significant due to its repetitive occurrence during the stance phase of the gait cycle. The knee joint experiences two to three times an individual’s body weight during each step of the gait cycle, which can contribute to wear injuries throughout life [43]. Then, during the stance phase, the knee supports the maximum body weight while being in a CPP. Compared to the knee, the shoulder joint reaches a CPP when the arm is overhead, with the glenohumeral joint abducted and externally rotated. While this position can occur naturally, such as during overhead reaching, this position is far less common compared to knee extension. The different biomechanical forces experienced in these joints during everyday tasks may explain the suspected lower incidence of shoulder OA compared to knee OA. As mentioned previously, the non-weight-bearing joints of the upper extremity can still develop OA as a result of overuse or injury. The use of any joint in its CPP will increase the forces that the joint experiences. It is the prolonged application of these forces that may predispose that joint to OA. Extended time in a CPP with the application of force can cause micro-traumas to the joint surfaces that may create sites that would allow OA to propagate. The knee has an anatomical advantage in its “screw-home” motion at maximum extension, which locks the knee and prevents movement of the joint in its CPP. The shoulder joint has no protective mechanism and can still experience force in its maximum CPP. It should be noted that OA has a variable presentation and is likely multifactorial; therefore, the true prevalence of GHOA is also likely related to the criteria for diagnosis—the diagnostic pathway.

### 3.3. Diagnosis

For each joint, there are unique diagnostic criteria that aid in the diagnosis of disease. The diagnosis of any form of osteoarthritis involves painting a full clinical picture, with history, physical exam, and diagnostic testing. Certain useful history items involve patient-reported outcome measurements (PROMs) such as the Quick Disabilities of the Arm, Shoulder, and Hand Questionnaire (QuickDASH), the American Shoulder and Elbow Surgeons shoulder score (ASES), and the Shoulder Pain and Disability Index (SPADI). These are questionnaires that can be used to screen for shoulder pain and disability and also measure outcomes after shoulder procedures are performed. Physical exams can include palpation for tenderness, testing the passive and active range of motion, and various special tests to screen for and rule out other shoulder pathologies. The most common diagnostic testing to be conducted is standard radiographic imaging, but computed tomography or magnetic resonance imaging can also be performed. The authors have chosen to focus on discussing the radiographic diagnostic criteria for GHOA.

Unlike OA of the knee, hip, or hand, which all have well-defined radiographic diagnostic pathways [54,55,56], GHOA does not have an agreed-upon criteria for diagnosis, and because of this, accurately defining GHOA is difficult [10,29,57,58]. Currently, diagnosis of GHOA relies on assessing symptomatic pain and radiological evidence of joint damage, such as glenohumeral joint space narrowing, osteophyte formation, cyst formation, and subchondral sclerosis [5] (Figure 5). These criteria imply that the diagnosis of GHOA is only possible in the advanced stages of the disease.

In addition to lacking a diagnostic pathway, there are currently six different radiological classifications to aid in a single diagnosis of GHOA. The classification systems include Samilson and Prieto, Allain, Gerber, Weinstein, Guyette, and Kellgren and Lawrence (Table 1). Each classification system uses variable grading and staging criteria such that no two classification systems align. For example, the description for staging, “narrowing of joint space”, is not consistent across the various classification systems. Nonetheless, the Samilson and Prieto classification is the most implemented for GHOA, consisting of 3 stages: mild, moderate, and severe. In an anteroposterior radiograph, the stages are classified primarily based on the size of osteophytes present on the inferior humerus or glenoid. Osteophytes measuring less than 3 mm define the mild stage, whereas osteophytes ranging from 3 mm to 7 mm accompanied by glenohumeral joint irregularity characterize the moderate stage. Lastly, osteophytes greater than 7 mm with subsequent narrowing of the glenohumeral joint and sclerosis characterize the severe stage of the disease. Interestingly, the World Health Organization (WHO) advocates for the Kellgran and Lawrence classification for epidemiological studies of OA, and while the classification has been used to describe GHOA, the original study by Kellgren and Lawrence did not include the glenohumeral joint [32,33,34] (Table 1). Once more, determining the actual prevalence of GHOA becomes elusive when there is no consensus on a single diagnostic criterion. As a result, due to an incomplete grasp of the actual impact on the population, there is a risk of delayed treatment or inadequate research to drive more effective treatments.

### 3.4. Treatments

Osteoarthritis is non-discriminatory and can affect different joints across the body in an asymmetrical fashion. Currently, there are limited treatment options and no curative therapies. Treatment options vary, spanning from conservative approaches like weight loss and non-steroidal anti-inflammatories (NSAIDs) to procedures such as joint injections (e.g., Platelet Rich Plasma (PRP), Hyaluronic Acid (HA), and corticosteroids) [59,60,61,62] and surgery [6]. The American College of Rheumatology and the Osteoarthritis Research Society International (OARSI) recommend conservative treatment, including walking, muscle strengthening, aerobic training, and neuromuscular training for patients with OA [63,64]. In conjunction with other conservative methods, supervised and progressive physical therapy is another treatment modality despite limited clinical evidence [19,65]. Moreover, although alternative therapies like acupuncture, cupping, dry needling, cannabidiol oil, platelet-rich plasma injections, and thermal therapy are utilized as conservative therapies for GHOA, the lack of scientific validation concerning their effectiveness has led the American Academy of Orthopedic Surgeons (AAOS) to refrain from endorsing their utilization in the treatment of GHOA [18]. Further research is needed on these alternative therapies to determine their effectiveness against OA.

If exercise and weight loss are ineffective, oral medications are available. NSAIDs are considered first-line for the conservative treatment of OA; however, there is a lack of clinical evidence to support their effective use in GHOA [10,18,63,66]. Alternatively, acetaminophen is an option for patients with contraindications to NSAIDs, but recent studies on acetaminophen demonstrated minimal to no effectiveness for their use in OA once again [63]. Another class of oral medications to consider is opioids, but these medications are not commonly prescribed for GHOA; given the side effects and the risk of dependence, both the AAOS and OARSI advise against the use of opioids as long-term treatment options for OA [18,63,64].

When conservative treatments show ineffectiveness, the subsequent treatment step involves intra-articular (IA) joint injections, usually employing glucocorticoids or hyaluronic acid. IA glucocorticoids are shown to be beneficial for the short-term management of both knee and hip OA, but there are insufficient data to support their use in GHOA [63]. In addition to glucocorticoid injections, there is mixed evidence on the use of hyaluronic acid injections in GHOA—a randomized control study by Tortato et al. demonstrated a statistically significant reduction in pain scores using IA hyaluronic acid injections [67]. Although studies demonstrate a potential benefit of IA hyaluronic acid injections, the AAOS strongly advises against the utilization of hyaluronic acid in the treatment of GHOA based on compelling evidence indicating its lack of benefit [18].

As a last resort, surgical interventions may be contemplated when conservative measures and joint injections have proven ineffective. There are two main surgical approaches: non-arthroplasty and arthroplasty. The non-arthroplasty procedures include arthroscopic and resurfacing. Arthroscopic procedures, such as arthroscopic debridement, are generally favored in younger, more active patients. Millett et al. described a comprehensive arthroscopic management technique that demonstrated an 85% survival rate after two years in young, active patients with advanced GHOA [68]. The other non-arthroplasty option—glenohumeral resurfacing—is also suitable for younger patients. This procedure maintains an individual’s natural anatomy and permits the potential for an elective total shoulder arthroplasty (TSA) later in life. For example, Peebles et al. documented a case of a younger patient (less than 35 years old) who underwent glenohumeral resurfacing after suffering from end-stage OA, with great success [69]. The second main surgical treatment approach for GHOA is arthroplasty. There are three main options when considering shoulder arthroplasty: anatomical TSA, hemiarthroplasty, or reverse TSA. The AAOS recommends an anatomical TSA over hemiarthroplasty because the anatomical TSA has demonstrated more positive outcomes in pain reduction and functionality [18]. However, patients with rotator cuff pathology or a bi-concave glenoid typically do better with a reverse TSA [17]. Largely, surgical treatment is appropriate after all other treatments have proven ineffective for the individual with end-stage GHOA.

The absence of evidence-based treatment modalities and curative options underscores the need for additional research in OA overall, as well as a focus on the glenohumeral joint itself. Current research is exploring various avenues for treating OA, including a deeper understanding of its pathogenesis, innovating methods for earlier detection, and bolstering more evidence for existing conservative treatments, as well as exploring alternative approaches.

Current research on the pathogenesis of OA is assessing multiple avenues, such as microenvironmental changes and metabolic shifts seen in affected joint tissues. For example, Zheng et al. in 2021 highlighted shifts in the metabolic pathways of chondrocytes and synoviocytes of OA joints, emphasizing the importance of investigating the field of immunometabolism and altered metabolic pathways as the key to understanding OA disease pathogenesis and unveiling potential therapeutic targets [15].

Additionally, efforts to identify early detection methods are gaining momentum in the literature. Current research includes assessing various data points, including biomarkers, radiographic data, histological data, MRI data, and arthroscopic criteria. Currently, OA is detected after subchondral bone damage has occurred, already being an irreversible stage of the disease, while there is currently no reliable method for detecting OA at a reversible stage. Despite ongoing research on promising techniques such as cartilage texture maps via transport-based learning [70] and AI-supported optical biopsy [71], further research is warranted in this field.

Lastly, while various alternative treatments complement conservative treatment, the need for research promoting the validation of current conservative options in OA treatment is well-established and continues to be assessed. This ongoing research underscores the importance of continued research efforts to improve the management of OA.

## 4. Conclusions

Although the prevalence of OA differs depending on anatomical location, it is evident that OA is less common in the glenohumeral joint. Nevertheless, despite numerous shared risk factors such as age, gender, genetics, trauma, and lifestyle, the exact cause of this difference remains puzzling. An examination of the literature suggests that this incongruity could be attributed to the intricate anatomy of the shoulder, the biomechanics of the joint, the physiological role of the glenohumeral joint, or possibly the diagnostic complexities involved. It is likely a complex multifactorial problem.

Irrespective of the specific causes, the absence of a clear, universally acknowledged diagnostic protocol for GHOA results in underreporting and underscores the urgent need for dedicated research aimed at unraveling the complexities of GHOA. Unlike in other joints, diagnosing GHOA often relies on patients’ medical histories, physical examinations, and radiographic assessments, leading to late diagnosis. Diverse radiological classification systems with disparate findings compound these diagnostic challenges. Establishing a standard radiological classification system for GHOA is crucial for enhancing diagnostic precision. Such research will not only improve patient outcomes but also pave the way for more effective treatments and preventive strategies tailored to tackling glenohumeral osteoarthritis.

## Figures and Tables

**Figure 1 jcm-13-02341-f001:**
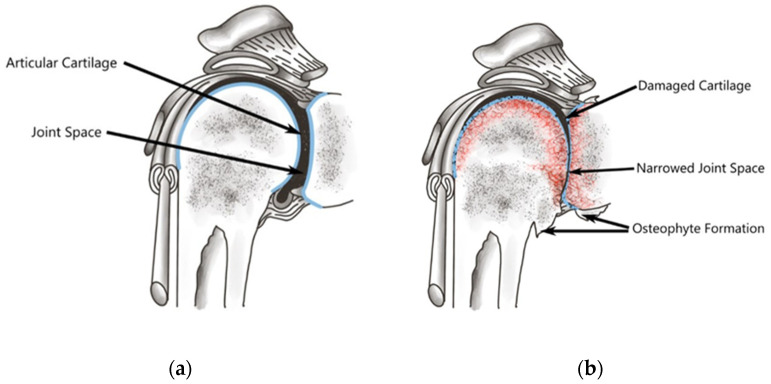
(**a**) A healthy shoulder joint demonstrating normal cartilage and structures and (**b**) an osteoarthritic shoulder joint over time compared to a healthy shoulder. Independently created by Hannah L. Wong.

**Figure 2 jcm-13-02341-f002:**
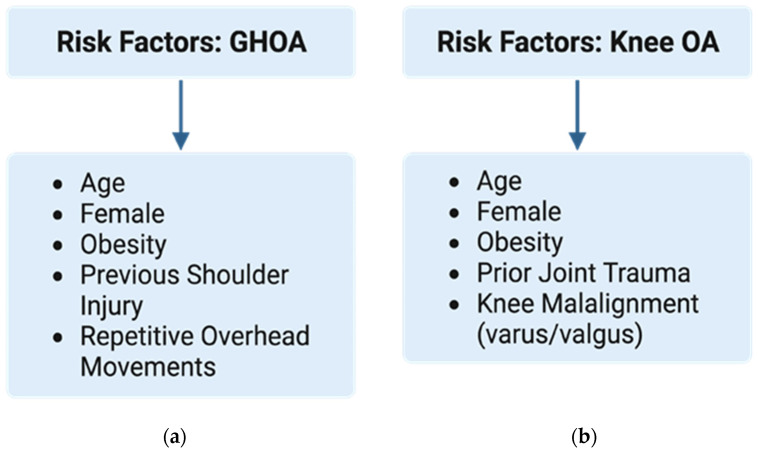
(**a**) Risk factors for glenohumeral OA and (**b**) risk factors for knee OA. Created with BioRender.com.

**Figure 3 jcm-13-02341-f003:**
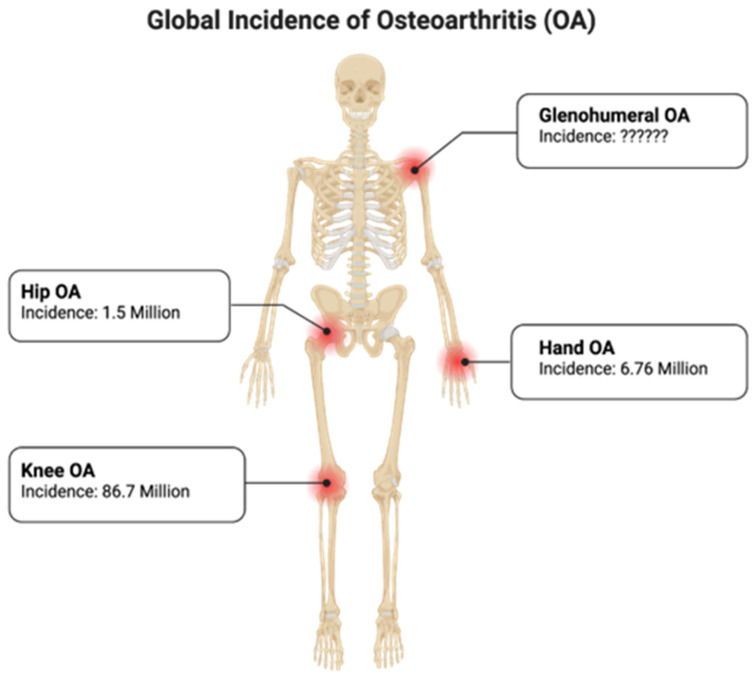
Global incidence of osteoarthritis in the knee, hip, hand, and shoulder joints in the United States. Created with BioRender.com.

**Figure 4 jcm-13-02341-f004:**
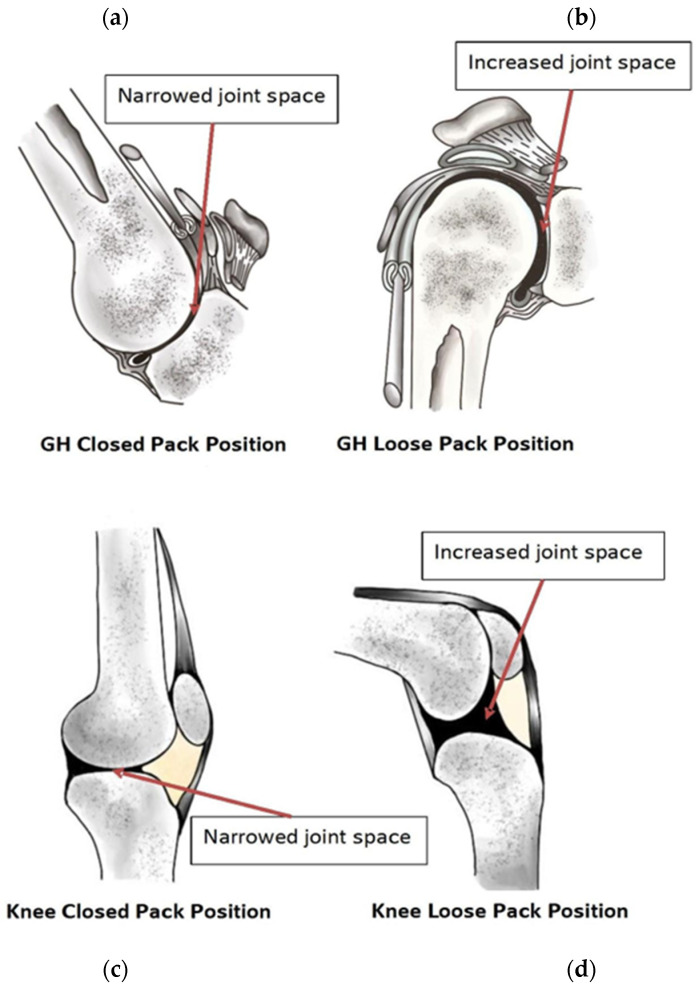
The glenohumeral joint in (**a**) a CPP and in (**b**) an MLPP and the knee joint in (**c**) a CPP and in (**d**) an MLPP. Independently created by Hannah L. Wong.

**Figure 5 jcm-13-02341-f005:**
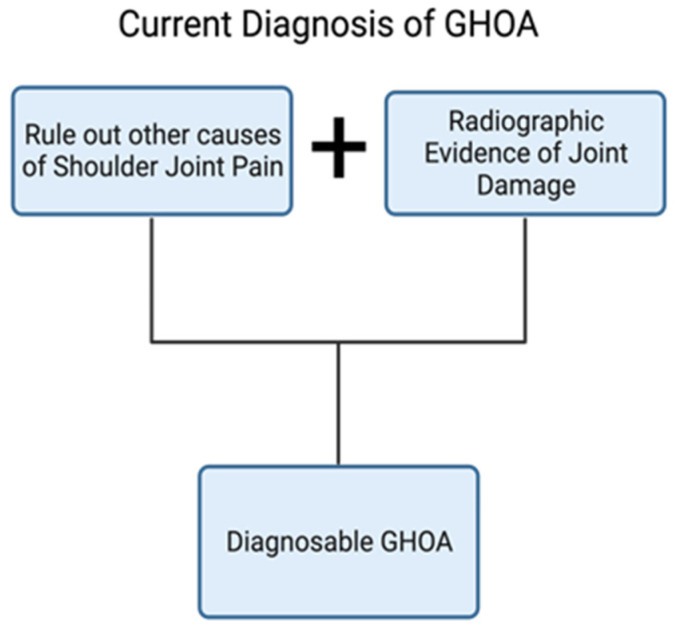
Current GHOA diagnosis flowchart. Created with BioRender.com.

**Table 1 jcm-13-02341-t001:** Six radiographic classification systems are used for categorizing GHOA. Adapted and used with permission from Elsharkawki et al. [32].

Radiographic Classification System	Grade/Stage	Description
Samilson and Prieto (SP) (Established 1983)	1	Inferior humeral or glenoid osteophyte <3 mm in height
	2	Inferior humeral or glenoid osteophyte between 3 and 7 mm in height, with joint irregularities
	3	Inferior humeral or glenoid osteophyte >7 mm in height with sclerosis and narrowing of joint space
Allain (Established 1998; modification of SP)	1	Inferior humeral osteophyte 1–3 mm in height
	2	Inferior humeral osteophyte 4–7 mm in height
	3	Inferior humeral osteophyte >7 mm in height
	4	Narrowing of joint space and sclerosis
Gerber (Established 1992; modification of SP)	1	Humeral head or glenoid osteophyte <3 mm in height
	2	Humeral head or glenoid osteophyte 3–5 mm in height, with joint line irregularity and subchondral sclerosis
	3	Degenerative changes to joint greater than previous grades
Weinstein (Established 2000)	1	Diagnosis during arthroscopy with no gross radiographic change
	2	Minimal narrowing of joint space with concentric humeral head and glenoid
	3	Moderate narrowing of joint space with early osteophyte formation
	4	Severe narrowing of joint space with osteophyte formation and loss of concentricity of humeral head and glenoid
Guyette (Established 2002)	0	No signs of osteoarthritis
	1	Mild sclerosis and/or osteophyte <2 mm on one side of the joint
	2	Narrowed joint space, large marginal osteophyte or multiple osteophytes, and/or presence of cysts
	3	Destruction of joint surface, bone-on-bone joint space narrowing, and/or presence of loose bodies
Kellgren and Lawrence (Established 1957)	0	Normal: no signs of osteoarthritis
	1	Doubtful: insignificant osteophytes with no joint space narrowing
	2	Minimal: definitive presence of osteophytes of minimal severity
	3	Moderate: narrowing of joint space with subchondral sclerosis.
	4	Severe: significant joint space narrowing, deformity of bone ends, and severe sclerosis

## Data Availability

Not applicable.

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
