# Peer review of "Glenohumeral Osteoarthritis: A Biological Advantage or a Missed Diagnosis?"

_jcm, 2024, doi:10.3390/jcm13082341_

Round 1
Reviewer 1 Report
Comments and Suggestions for Authors
- figure 1 should be replaced with a representation that is more scientific and related to glenohumeral joint, and not the knee; at current state, it is a representation for a 12th grade student, which is way too general. I suggest the authors emphasize on more and granular features of OA, that are relevant in 2024.
- Clarify that the prevalence rate can vary significantly among different subpopulations, and these figures presented are an estimate
- the incidence of glenohumeral is known, and it is 85 and 94% in men and women over the age of 80 years (https://pubmed.ncbi.nlm.nih.gov/37674967/). Please clarify
- it would be more accurate to mention that findings from one country might not be universally applicable
- highlight that the socioeconomic impact of OA can vary widely
- specify particular areas where research is lacking, such as: a. understanding the pathogenesis of OA, b. developing new treatment modalities, c. identifying early detection methods
- "rotate cuff muscles" should be "rotator cuff muscles."
- clarify how both weight-bearing and non-weight-bearing joints are susceptible to OA for different reason
- include information on how changes in synovial fluid composition and cartilage degradation contribute to OA
Author Response
Hello,
Thank you so much for your constructive feedback. The authors have addressed your comments; we look forward to your thoughts on our changes. Have a wonderful day.
Thank you again!
Reviewer 2 Report
Comments and Suggestions for Authors
Analyzing the submitted article "Glenohumeral Osteoarthritis: A Biological Advantage or a Missed Diagnosis?", the reader feels like reading a chapter of a book. The paper lacks the characteristic structure of a systematic review or a narrative review according to PRISMA guidelines. The study is divided into two chapters: Introduction and Discussion, and Conclusion. There is no information regarding the methodology or the selection criteria for the analyzed literature. The authors do not mention functional assessment tools such as core sets according to ICF or the QuickDASH questionnaire in describing the diagnosis based solely on X-ray diagnostics, without considering other indicators of inflammation like ESR or ultrasonographic examination. The authors cite literature from 1957, 1983, and 1990 – certainly more current literature on this topic is available. The article appears to be a collection of information not thoroughly considered by the authors.
Author Response
Hello,
Your feedback was taken into account and the authors have addressed your comments. Thank you for your time and we look forward to hearing your thoughts as we move forward in the peer review process.
Thank you again. Have a wonderful day.
Reviewer 3 Report
Comments and Suggestions for Authors
Dear authors,
Thank you for submitting your manuscript titled “Glenohumeral Osteoarthritis: A Biological Advantage or a Missed Diagnosis?” for review.
The manuscript is a synthesis whose subject is glenohumeral osteoarthritis. Overall, the article is well written, but it is deficient from a scientific point of view. Detailed and recent knowledge is needed in the subsections of anatomy, diagnosis and treatment. The conclusions are actually summaries of the ideas presented in the other chapters. The bibliography contains few titles for a review article. This review does not provide new scientific data or analytical correlations.
In addition, I have identified few areas where the manuscript could benefit from further enhancements. Below are my detailed suggestions:
- Line 83 – indicate the sources that mention GHOA as the third most common type
- Lines 85, 323, 332, 338 – it is recommended to place the reference after the authors' names. Ansok and Muh [7]
- Line 182-183 - there are no medial and lateral cruciate ligaments. Probably - tibial and fibular collateral ligaments
- Lines 200 and 203 – Figure 4 - specify which of the 4 images (Figure 4 a,c)
- Lines 250-251 – the radiological evidences characterize all stages. See Samilson and Prieto classification.
- Lines 351-352 - the statement is not a conclusion of this study but is demonstrated by Long at al. [6]
- Lines 352-353 – this conclusion is not supported by a present clinical study.
- Lines 355-357 – same question like the title. I suggest commenting on this question with arguments for and against in the discussion chapter.
- Lines 358-365 - recapitulation of ideas already presented. They are not conclusions of the present study.
I hope these suggestions will be helpful in strengthening your manuscript and better conveying the research you have undertaken. Overall, my peer review is a serious major revision.
Looking forward to seeing the revised version of your work.
Best regards
Author Response
Hello,
Thank you very much for taking the time to read and provide feedback on our manuscript. The authors have addressed your comments and we look forward to hearing your thoughts. Have a wonderful day.
Thank you again.
Round 2
Reviewer 1 Report
Comments and Suggestions for Authors
Authors have addressed my previous requests. The manuscript looks scientifically correct now.
Author Response
Thank you for your kind review and comments back.
Reviewer 3 Report
Comments and Suggestions for Authors
Dear authors,
Thank you for resubmitting your manuscript titled “Glenohumeral Osteoarthritis: A Biological Advantage or a Missed Diagnosis?”
I appreciate the efforts of the authors to improve the manuscript. From a scientific point of view, improvements are observed in all chapters, except Conclusions. However, some aspects has to be improved. I will precisely specify these aspects:
- For an eloquent review regarding a certain pathology, a search period of 8 months is much too short. Moreover, only 6 titles from this period are included in the bibliography. Being a narrative review the search can be extended.
- You mentioned the search period in the Abstract but not in the Materials and Methods.
- The conclusions must be just statements that emerge from the conducted study. They have nothing to look for references and discussions in the Conclusions. You draw the conclusions of your study; do not paraphrase the conclusions of other papers.
- references 8,9 and 13 are incomplete (the authors are not listed)
My peer review is a minor revision.
As a recommendation:
The authors are a team and all are responsible for the way the review is carried out. The reviewers are not interested in which author formulated the answer. You must respond to each author specifically to his requests, without mixing them with those of another reviewer. In conclusion, it is necessary to have a Cover Letter of response for each individual reviewer. All changes in the text should be marked with the same colour - read the Instructions for Authors before submit.
Author Response
Thank you for the time you took to provide meaningful feedback to our manuscript. We have updated the manuscript to reflect changes in regard to your feedback. The cover letter specifically outlines our changes, as well as the changes can be seen highlighted in the manuscript itself.